# Consecutive Shifts: A Repeated Measure Study to Evaluate Stress, Biomarkers, Social Support, and Fatigue in Medical/Surgical Nurses

**DOI:** 10.3390/bs13070571

**Published:** 2023-07-10

**Authors:** Mona Cockerham, Duck-Hee Kang, Margaret E. Beier

**Affiliations:** 1School of Nursing, Sam Houston State University, The Woodlands, TX 77380, USA; 2Houston Methodist, Willowbrook Hospital, Houston, TX 77030, USA; 3Cizik School of Nursing, University of Texas Health Science Center, Houston, TX 77355, USA; 4Department of Psychological Sciences, Rice University, Houston, TX 77005, USA; beier@rice.edu

**Keywords:** consecutive shifts, nurse, stress, fatigue, biomarkers

## Abstract

Nurses report that they are required to work during their scheduled breaks and generally experience extended work times and heavy workloads due to staffing shortages. This study aimed to examine changes in personal, work-related, and overall stress, as well as biological responses and fatigue experienced by nurses during three consecutive 12 h workdays (i.e., the typical “three-twelves” schedule). We also considered the moderating effects of social resources. This prospective study of 81 medical/surgical nurses who completed questionnaires and provided saliva samples at four designated intervals (i.e., pre-shift and post-shift on workdays 1 and 3). Fatigue reported by night shift nurses increased significantly over three consecutive workdays (*p* = 0.001). Day shift nurses said they encountered more social support than those on the night shift (*p* = 0.05). Social support moderated the relationship between work-related stress at baseline and reported fatigue on day 3.

## 1. Introduction

Fatigue in the nursing workforce directly impacts patient safety and health outcomes, as well as economic consequences that are largely due to the cumulative effects of consecutive work shifts [1]. Since the 1990s, consecutive shift scheduling has become a common practice in hospitals across the United States (US). Commuting and unremunerated work time (e.g., professional committees and continuing education) can expose nurses to workdays that are substantially longer than 12 h [2]. Furthermore, household responsibilities (e.g., managing childcare, cleaning, shopping and preparing meals, and supervising homework) reduce the time available for sleep [3]. Nurses frequently highlight the benefits of consecutive 12 h shifts because they allow for extended time away from work [4]. However, current research does not strongly support the benefits associated with consecutive shifts [2].

Furthermore, consecutive shift work may impact circadian rhythms, including those modulating physical recovery, cortisol secretion patterns, and personal fatigue. Everyday home and work demands can lead to increased stress levels and fatigue; collectively, these stressors may have a negative influence on health and, left unchecked, may ultimately lead to chronic fatigue [3,5,6,7,8,9]. Social resources can help moderate the effects of stress on nurses through, for example, a supportive family, meaningful friendships, and a supportive work environment [10,11].

There is little research on the impact of more than two consecutive work shifts on stress, biomarkers, and fatigue experienced by hospital-based nurses in the US. Among the studies that have been published, many include relatively small sample sizes, and most studies are in Asian countries with rotating shifts, which is not common practice in the US [7,8,12]. Moreover, the US does not provide its citizens with universal government-funded healthcare similar to Asian or European countries, except in care of the elderly (Medicare) and disabled or indigent (Medicaid); thus, in many hospital settings, a large percentage of patients are privately insured. Many Americans maintain strong opinions and values regarding healthcare and its delivery structure and expect excellent care [13]. Due to these differences in the healthcare structure and funding in the US, the current study reports the response of US nurses’ relationships to stress, biomarkers, and fatigue across consecutive shifts [14].

This study aims to determine the impact of consecutive 12 h shifts using predictors of self-reported stress, biomarkers, and an outcome measure of fatigue. A sample of nurses working three consecutive day or night shifts provided biomarkers and self-report measures to assess stress and fatigue. These are critical issues in need of significant attention by policymakers in the US: As many nurses are nearing retirement age, attention to these concerns may ultimately alleviate the anticipated shortage of hospital-based nurses.

### 1.1. Literature Review

A search of PubMed, Psych Info, Web of Science, CINAHL, MEDLINE, Psychology, and EBSCO Information Systems using the terms “shiftwork OR shift work” AND “consecutive” AND “nurs*” from 2019 to 2022 yielded only a few primarily simulation-type studies. These findings revealed a gap in the literature as we could find little information on the real-world impact of 12 h consecutive shifts on nurses in the US. The research we found showed that nurses who work consecutive shifts need to have an active plan for recovery to restore their physical and emotional health and to prevent illnesses that might result in the need for sick leave [15]. Night shift nurses can experience long-term fatigue due to interruptions in their circadian rhythms unless a similar schedule is maintained on off days, and it is recommended that nurses should be provided with at least three days off after working two consecutive night shifts [16,17]. Similarly, results from a repeated measures study revealed that nurses experienced significant fatigue levels over two 12 h work shifts [18]. Another study in New Zealand reported the cumulative effects of three 12 h shifts, including a significant decline in muscle function after a single 12 h shift. A simulation study recorded the results of 14 continuous days of activity in which standing and walking activities were compared during a 12 h day shift and night shifts [19]. Among the results, nurses walked less during the third consecutive night shift and sat more during the second and third consecutive night shifts compared to day shift nurses. Stress, fatigue, and cortisol levels increased significantly from baseline to day 2 in this study [20]. The current study was interested in whether stress and fatigue increased over consecutive shifts. Thus, we compared levels of stress from day 1 to day 3 shifts.

**Hypothesis** **1.***Perceived stress, biological stress responses, and fatigue will be significantly higher on day 3 than on day 1.* 

Studies have also examined differences between day and night nurses. As part of a simulation study, day and night shift nurses performed similarly on cognitive tasks even over time [21]. However, night shift nurses reported a small but statistically significant decrease in communication skills. Another study reported increases in fatigue over two 12 h shifts, with slightly more significant fatigue reported by night shift nurses; in this study, the differences between day and night shifts accounted for a significant portion of the variance [22]. Collectively, the results of these studies suggest that night shift nurses are more likely to develop significant fatigue than day shift nurses. The goal of the current study was not to examine differences between day and night nurses, however. As such, we do not have a specific hypothesis about them, but examine them as part of our analysis. 

### 1.2. Stress

Stress is defined as a non-specific response of the body to any demand [23]. Stress can be a response to demands, people, or events that leave an individual feeling unable to meet actual or perceived expectations [23]. In the current study, four types of stress were measured: perceived personal, work-related, general, and nurse-specific stress. We also measured stress using biomarkers, which are markers of known neuroendocrine responses resulting from the activation of the hypothalamus-pituitary-adrenal axis (HPA) [16]. We measured cortisol, the primary stress hormone that promotes the release of stored energy during times of stress, and which reflects long-term responses to stress. By contrast, alpha-amylase levels were used to measure acute stress. 

**Hypothesis** **2.***High levels of personal and work-related stress measured on day 1 will predict high biological stress responses on day 3.* 

### 1.3. Fatigue

Fatigue is exacerbated by heavy workloads and reductions in productivity that lead to care left undone and can negatively affect patient outcomes [7]. Fatigue also leads to slow cognition, errors in practice, musculoskeletal injuries, long-term poor health outcomes for nurses, sickness-related absenteeism, and inferior organizational outcomes when the costs associated with inadequate care are perceived as lost revenue [21,24,25,26,27,28,29]. Decreased alertness and sleepiness are among the most common outcomes of fatigue [30]. Furthermore, fatigue influences a nurse’s physical, mental, and emotional well-being, leading to poor decision-making and relationship conflict [21]. Fatigue is related to turnover intentions among nurses, which bears a heavy cost to both employers and society [30]. In this study, we were interested in examining if baseline levels of stress would predict fatigue at the end of three consecutive shifts.

**Hypothesis** **3.***High levels of personal and work-related stress detected on day 1 will predict high levels of fatigue on day 3.* 

### 1.4. Social Resources

Social resources are mechanisms that can assist individuals with efforts to cope and reduce pathological responses to stress [10]. A published conceptual analysis of social support revealed a positive relationship between health and social support, including emotional, informational, instructional, and appraisal attributes [31]. Among its key components, social support for nurses should provide insight into effective coping behaviors and recognition of self-worth, and include positive social networks and a social workplace climate. A supportive hospital environment can provide adequate organizational support that includes resilience intervention resources designed to help nurses meet job expectations and handle work-related stress in a positive light [32,33]. Adequate social support can reduce these risks [10,24]. As such, we were interested in whether social resources would moderate the relationship between stress and fatigue.

**Hypothesis** **4.***The availability and use of social resources measured at baseline will moderate the relationship between work-related stress at baseline and fatigue on day 3.* 

## 2. Methods

### 2.1. Participants and Procedures

This study was a prospective, within-subjects repeated design that enrolled nurses from five medical/surgical units who worked 12 h shifts on three consecutive days or nights from January to April 2016. One-time surveys 24–36 h before a consecutive shift pattern start with a Likert scale assessment of stress and fatigue along with each biomarker collection at the beginning and end of each shift (Table 1).

### 2.2. Sample Selection

The sample size was computed a priori using a G*Power 3.1 [34] for paired *t*-tests considering a small effect size (Cohen’s *d* = 0.34). These values were estimated based on findings reported in a pilot study [20]. At α = 0.05 and Cohen’s *d* = 0.34, a paired *t*-test will have 80% power using a sample size of 71 [34].

### 2.3. Measures

Nurses provided their age (in years), gender, ethnicity, hours worked, overtime during the study, education level, years of experience, and practice indicators. The Perceived Stress Scale (PSS, α = 0.80) was used to measure baseline stress [35]. Work-related stress was measured at baseline with the Nurse Stress Scale (NSS, α = 0.85) [36]. The Visual Analog Scale-Stress (VAS-S, scores 0–10) was administered before and after work on day 1 and day 3 to assess pre-and post-shift stress. The Multidimensional Fatigue Inventory (MFI-20, α = 0.80) was used to measure fatigue at baseline [37]. Repeated fatigue levels were evaluated both pre-and post-shift using a single-item Visual Analog Scale-Fatigue (VAS-F, scores 0–10). The 12-item Multidimensional Scale of Perceived Social Support was used to assess social support (α = 0.95) [36]. All reliability estimates were derived from the current data collection.

Biomarkers were used to measure objective stress, including salivary cortisol as a biomarker of HPA activity and salivary α-amylase as a surrogate measure of the activity of the sympathetic nervous system [16]. Changes in cortisol levels over a 12 h shift measured in day shift nurses were expected due to physiological circadian rhythms. Nurses were instructed to abstain from eating, smoking, brushing their teeth, and drinking sweetened or caffeinated beverages 60 min before the saliva collection. Saliva samples were collected four times pre-shift (2X) and post-shift (2X) on days 1 and 3 of the scheduled three-day 12 h workdays. Specimens were stored in a refrigerator until transported to the lab. Specimens were batch-assayed using Salimetrics Enzyme Immunoassay kit instructions (Salimetrics, LLC.). The intra-assay coefficient of variation was <10%, and the intra-assay coefficient of variation was <15% for cortisol.

Fatigue was measured as the outcome variable of this study. The Multidimensional Fatigue Inventory (MFI-20) measures general fatigue, physical fatigue, mental fatigue, reduced activity, and reduced motivation [37]. The reliability estimate is α = 0.80. We assessed repeated fatigue levels using a visual analog scale for fatigue (VAS-F), with scores ranging from 0 to 10.

### 2.4. Data Analysis

Descriptive statistics included means, standard deviations, frequencies, and percentages. Data were examined for missing values and normal distribution using scatter plots and histograms. Salivary cortisol and α-amylase levels were not normally distributed and were log-transformed. Within-person comparisons were made using paired-*t*-tests. Differences between day shift and night shift nurses were assessed using Chi-square tests for categorical variables and independent *t*-tests for continuous variables. Correlation and hierarchical regression methods were used to assess hypotheses 2, 3, and 4. Analyses were performed using STATA version 16.1.

## 3. Results

Table 2 shows demographic information for the study population. The study cohort included 43 day shift nurses (53%), and 38 night shift nurses (46%). The majority of ages were within two group ranges: 30–39 (25%), and 40–49 (43%), with the most common race being Asians (38%), Caucasians (32%), African Americans (16%) and lastly Hispanics (10%). The majority were BSN graduates (85%) with years of experience groups from 0 to 5 years (25%), 6–10 years (25%), and 11–15 years (30%). Most reported working an average of 12.5 h per shift (SD ± 0.5). The majority (78%) of the nurses were married and reported that spouses helped with household management (77%). Comparisons of baseline measures obtained from the day and night shift nurses are also shown at the bottom of Table 2. Although day nurses reported that they experienced higher levels of social support at a baseline than were reported by night shift nurses (*p*-0.05, *d* = 0.46), there were no significant differences between groups on the other variables.

### 3.1. Hypothesis 1: Stress, Biomarkers, and Fatigue Increase over Time

Within-person comparisons for day and night shift nurses from day 1 to day 3 are shown in Table 3. We observed no significant differences in stress as measured by VAS-S scores either within or between shifts. While fatigue (as measured by VAS-F) increased from day 1 to day 3, these differences achieved statistical significance only among the night shift nurses (*p* = 0.001, *d* = 0.67). When comparing outcomes between day and night shift nurses, there were no significant changes in salivary α-amylase levels within or across shifts. However, we identified variations in normal circadian cortisol levels, specifically among the day shift nurses (*p* = 0.007). These findings provide mixed support for Hypothesis 1, which was that perceived stress, fatigue, and biological stress responses would be higher on day 3 than on day 1 (Table 3).

### 3.2. Hypothesis 2: Relationships between Baseline Predictors of Personal and Work-Related Stress and Levels of Cortisol and α-Amylase

Correlations among self-report assessments and biomarkers are shown in Table 4.

Hierarchical regression analysis was performed separately for day and night nurses to determine the relationship between personal (PSS) or work-related (NSS) stress at baseline and post-shift salivary cortisol levels on day 3 after controlling for covariates (i.e., race, marital status, care for extended family, years in nursing, and type of shift). Data collected from the day and night nurses were evaluated separately because of the differences in cortisol detected between these two groups. Cortisol is a continuous measurement, and linear regression was used for this analysis. Personal and work-related stress were not significant predictors of post-shift cortisol or α-amylase levels on day 3 for either day or night shift nurses (Table 5). Thus, our findings provided no support for Hypothesis 2. There were no other significant differences in the patterns for day and night shift nurses; therefore, the data for these two groups were combined to test hypotheses 3 and 4.

### 3.3. Hypothesis 3: Relationship between Baseline Personal and Work-Related Stress and Fatigue

To test Hypothesis 3 (i.e., that high levels of personal and work-related stress on day 1 would predict high levels of post-shift fatigue on day 3), we combined data collected from both day and night shift nurses in a hierarchical regression model using day 3 post-shift fatigue as the outcome controlling for the same covariates as listed above. Personal and work-related stress did not account for significant variance in post-shift fatigue (VAS-F) on day 3, providing no support for Hypothesis 3 (Table 6).

In addition to examining personal and work-related stress, we performed a hierarchical regression to determine whether overall stress (VAS-S at baseline) predicted post-shift fatigue on Day 3 after accounting for the covariates and found that overall stress (VAS-S) explained a significant change in variance accounted for (*R*^2^ = 0.08; *p =* 0.008) above the covariates (Table 6). Collectively, these findings suggest that overall baseline stress assessed by VAS-S (1–10 scale) was a superior predictor of post-shift fatigue compared to the PSS and NSS.

### 3.4. Hypothesis 4: Moderation between Stress and Fatigue

To examine Hypothesis 4, a hierarchical regression analysis was performed to evaluate the interaction between work-related stress (NSS scores) and social resources (as measured by the Multidimensional Scale of Perceived Social Support [MSPSS]). For this analysis, a dummy variable was created for MSPSS based on the median value to delineate high (>55) versus low (< or equal to 55) scores, and the NSS measure was centered on avoiding collinearity problems. The main effects of covariates, NSS, and the MSPSS were entered first into the model; the interaction term between NSS and MSPSS was entered as the second step. Moderation was established by a significant change in R^2^ for the interaction after accounting for the main effects of stress, social support, and covariates. Our findings identified a significant interaction between NSS and MSPSS at baseline that predicted post-shift fatigue (VAS-F) on day 3 (β = 0.28; *p* = 0.035). As shown in Table 7 and Figure 1, there is a statistically significant inverse relationship between work stress (NSS) and post-shift fatigue (VAS-F) on day 3 (slope = −0.109; *p* = 0.042), specifically for nurses with fewer social resources (blue line). Surprisingly, fatigue declined as stress increased for these nurses. For nurses with more social resources, the relationship between NSS and post-shift fatigue on day 3 was not statistically significant (slope = 0.005; *p* = 0.615; discontinuous red line). In other words, when nurses reported social support, stress did not have a significant effect on fatigue. This provides support for hypothesis 4, which stated that social support would moderate the relationship between stress and day 3 fatigue, although the pattern of relationships was not expected.

## 4. Discussion

This study explored the impact of changes in stress, biological responses, and social resources on fatigue experienced by both day and night shift nurses who worked three consecutive 12 h hospital shifts in one of five different medical/surgical specialty units. The results reveal that fatigue increases over three consecutive shifts, particularly for night shift nurses, and that social support can mitigate these effects. We found that while fatigue reported by night shift nurses (i.e., VAS-F scores) increased across three consecutive shifts, there were no other significant changes in self-reported stress levels or biomarkers. Our findings align with another study that reported an increase in fatigue four hours into a shift and after two 12 h work shifts and cumulative fatigue across consecutive shifts [21]. This study provides an essential baseline for future studies given that healthcare is a 24 h profession, and consecutive 12 h workdays are a common scheduling practice in US hospitals. Relatively few studies have previously documented that long hours and increasingly complex acute patient care performed by nurses contribute to stress and fatigue [5,12].

Although our findings support using a VAS scale to assess stress and fatigue across shifts, it is not clear why other baseline assessments (PSS/NSS) were not sensitive to predicting increases in biological stress measures. This is particularly curious given that other studies reported increases in stress in situations involving rapid patient turnover, few staffing resources, and conflicting care initiatives [38]. One reason for this discrepancy may be that our methods were not sufficiently sensitive and were unable to capture changes in stress and fatigue over three shifts, particularly the assessment of biomarker levels. Another reason may be related to the fact that the nurses in our sample were not working overtime and thus may have been more resilient to levels of fatigue and stress that might otherwise accumulate over three consecutive shifts. Moreover, we did not evaluate any of the actual events that happened during a shift that might contribute specifically to higher levels of fatigue and stress. Future research might address these questions with more sensitive measures and assessments of specific shift activities.

Our results also suggest that social resources can protect against stress-related fatigue. Specifically, we found that stress did not impact fatigue levels for nurses reporting higher levels of social support. By contrast, those reporting limited access to social support were likely to experience less fatigue at the end of their shift on day 3 as work-related baseline stress increased. This pattern of relationships was not expected. It could be that stress may serve as an agitator for these nurses, reducing their perceived fatigue levels. Future research is needed to further understand these relationships.

Moreover, our results suggest that night shift nurses are less likely to report adequate access to social support than day shift nurses. These findings imply that supportive work relationships with colleagues and hospital leaders, which are more prevalent for day nurses, provide a buffer against the development of work-related stress. Other research directions might include an examination of interventions that promote organizational support for stress and fatigue [24,25]. Perceived organizational support is another social resource that might be explored in future research; employees who believe their organizations value their contributions and care about their well-being typically experience more positive outcomes [3].

### 4.1. Limitations

This study was limited to nurses employed in medical-surgical units of a single magnet community hospital. Among other limitations, cortisol levels may be influenced by factors such as sleep quality, coffee intake, and ovulation time. Similarly, our findings may not have captured the highest stress or fatigue levels; some nurses stated that they were aware that their stress levels were high and thus declined to participate (personal communication, Cockerham, 2016).

The relationship between demographic differences (i.e., gender, age, and years of experience) and fatigue and stress experienced when working overtime or consecutive 12 h shifts might also be explored. Efforts might be made to provide 8 h shifts for older nurses; this may permit them to extend their career time and continue to provide the benefits of their experience to those who are new to the field. Moreover, there are currently few studies that capture the impact of rapid return to work, more than three consecutive shifts, and seven days on—seven days off staffing patterns, all of which may have a negative impact on nurse well-being and patient safety [39]. Researchers should continue to evaluate the effects of fatigue on various hospital scheduling patterns and perform interventions designed to support the well-being and retention of nurses in the hospital setting. Additional studies will be needed to determine the impact of working consecutive 12 h shifts together with the adaptation of a healthy work environment methodology based on recommendations provided by Healthy Nurse and Healthy Nation [40]. It might also be helpful to replicate this study in settings where resources and quality of care do not currently meet the highest outcomes standards.

### 4.2. Theoretical Implications

Biobehavioral approaches facilitate the understanding of mind–body interactions on health [41]. Kang’s Expanded Biobehavioral Interaction model represents a holistic view of addressing complex biobehavioral interactions with health outcomes like fatigue. Fatigue is a complex outcome influenced by various factors, such as psychosocial, behavioral, individual, and environmental determinants, which influence nurses’ performance and ability to care for patients across multiple shifts into overtime more significantly than this study evaluated. Future studies should evaluate fatigue above 36 h or full-time, considering overtime, as well as further define fatigue as physical, mental, occupational, transient, or cumulative.

### 4.3. Administrative Implications

Our results suggest that nurses who perceive relatively more social support will report lower levels of fatigue, regardless of their work-related stress. This suggests that work environments that facilitate inter-professional collegial relationships will benefit nurses. Magnet hospitals are more likely to support fatigue countermeasure programs with social-support building practices [42].

## 5. Conclusions

Many nurses leave hospital practice and report stress as the underlying reason [38]. Furthermore, many nurses work two jobs and maintain schedules that are not determined by a single employer. While consecutive shifts are standard throughout the US, the impact of this scheduling practice on stress and fatigue experienced by nurses is not well understood. The results of this study reveal that fatigue increases over three consecutive shifts, particularly for night shift nurses, and that social support can mitigate these effects. Collectively, our findings provide important insight into directions for future research and the development of interventions designed to reduce fatigue. Researchers, hospital leadership, and professional organizations should continue to evaluate the effects of fatigue on nurses and partner with supervisory staff to support the integration of fatigue countermeasures in hospital settings [43].

## Figures and Tables

**Figure 1 behavsci-13-00571-f001:**
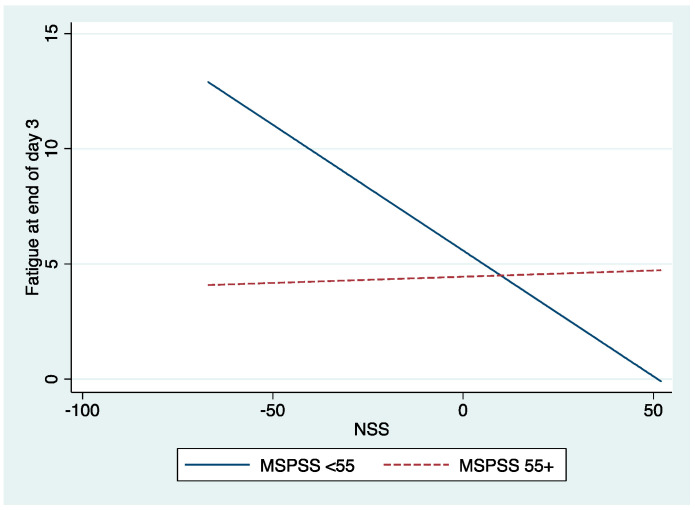
The interaction of work-related stress at Nurse Stress Scale (NSS) and Social Support (MSPSS) with post-shift fatigue (VAS-F) on Day 3.

**Table 1 behavsci-13-00571-t001:** Repeated measures data collection.

	Baseline	Day 1 Pre-Shift	Day 1 Postshift	Day 2	Day 3 Pre-Shift	Day 3 Postshift
Demographics	X			No data		
Personal Stress Scale (PSS)	X			collected		
Work Stress (NSS)	X					
Fatigue (MFI)	X					
Social Resources (MSPSS)	X					
Stress (VAS-S)	X	X	X		X	X
Fatigue (VAS-F)	X	X	X		X	X
Saliva Cortisol		X	X		X	X
Alpha Amylase						

**Table 2 behavsci-13-00571-t002:** Characteristics of the study population.

	All Nurses (N = 81)	Day Shift Nurses (n = 43)	Night Shift Nurses (n = 38)	Comparison
Variables	Mean (SD) or Frequency (%)	Mean (SD) or Frequency (%)	Mean (SD) or Frequency (%)	*p*-Value
Age (years)				
20–29	13 (16)	7 (9)	6 (7)	0.66
30–39	21 (25)	10 (11)	11 (14)	
40–49	35 (44)	19 (23)	16 (20)	
50–59	9 (11)	5 (6)	4 (5)	
60+	3 (4)	2 (3)	1 (1)	
Gender				
Male	18 (22)	10 (12)	8 (10)	0.51
Female	63 (78)	33 (41)	30 (37)	
Ethnicity				
Asian	31 (38)	13 (16)	17 (22)	0.26
Caucasian	26 (32)	15 (19)	11 (14)	
Hispanic	8 (10)	5 (6)	3 (4)	
Afr. Am.	13 (16)	7 (8)	7 (8)	
Other	3 (4)	3 (4)	0	
Education				
ADN	6 (7.5)	2 (2)	4 (5)	
BSN	69 (85)	37 (46)	32 (39)	
Graduate degree	6 (7.5)	4 (5)	2 (2)	
Years in Nursing				
0–5	20 (25)	12 (15)	8 (10)	0.93
6–10	20 (25)	9 (11)	11 (14)	
11–15	24 (30)	11 (14)	13 (16)	
16–20	14 (17)	9 (11)	5 (6)	
21–30	3 (3)	2 (2)	1 (1)	
Hours worked/week				
36–40	53 (65)	27 (33)	26 (32)	0.86
40–50	24 (30)	15 (19)	9 (11)	
50–60	4 (5)	1 (1)	3 (4)	
Hours worked/day	12.53			
12	41 (51)	22 (27)	19 (24)	0.90
13	38 (47)	20 (25)	18 (22)	
14	2 (2)	1 (1)	1 (1)	
Married				
Yes	63 (78)	34 (42)	29 (36)	0.67
No	18 (22)	9 (11)	9 (11)	
Care for extended family in the home				
Yes	21 (26)	13 (16)	8 (10)	
No	60 (74)	30 (37)	30 (37)	0.35
Spouse helps with household management				
Yes	62 (77)	33 (41)	29 (36)	
No	19 (24)	10 (12)	9 (11)	0.96
Baseline Measures				
Personal Stress (PSS)	18.7 (5.2)	18.0 (5.6)	19.5 (4.6)	0.21
Work-related Stress (NSS)	112.8 (28.3)	113.1 (29.4)	113.0 (27.4)	0.94
Overall Stress (VAS-S)	3.51 (2.8)	3.5 (2.8)	3.4 (3.0)	0.92
Multidimensional Fatigue (MFI)	45.1 (14.1)	45.5 (13.6)	45.5 (13.7)	0.80
Overall Fatigue (VAS-F)	3.54 (3.1)	3.7 (3.3)	3.4 (3.0)	0.74
Social Resources (MSPSS)	68.36 (17.26)	72.1 (11.9)	64.2 (21.2)	0.05

**Table 3 behavsci-13-00571-t003:** Comparison of day and night nurses’ variables.

	Day Nurses				Night Nurses		
Variables	Mean (SD)	Mean Diff	*p*		Mean (SD)	Mean Diff	*p*
Stress VAS							
Day 3 Post-shift Day 1 Pre-shift	4.2 (2.5) 4.0 (2.8)	0.2	0.68	Day 3 Post-shift Day 1 Pre-shift	3.1 (2.5) 3.2 (2.1)	0.1	0.87
Cortisol μg/dL							
Day 3 Pre-shift Day 1 Pre-shift	−1.65 (0.93) −1.20 (0.79)	−0.45	0.007	Day 3 Pre-shift Day 1 Pre-shift	−1.79 (0.81) −1.97 (1.02)	0.18	0.19
Day 3 Post-shift Day 1 Post-shift	−2.95 (1.01) −3.11 (0.96)	0.16	0.30	Day 3 Post-shift Day 1 Post-shift	−2.31 (1.37) −1.96 (0.83)	−0.33	0.12
α-Amylase μg/dL							
Day 3 Pre-shift Day 1 Pre-shift	4.22 (0.93) 4.16 (0.90)	0.06	0.59	Day 3 Pre-shift Day 1 Pre-shift	4.76 (0.79) 4.82 (0.92)	−0.06	0.54
Day 3 Post-shift Day 1 Post-shift	4.61 (0.80) 4.71 (0.81)	−0.10	0.43	Day 3 Post-shift Day 1 Post-shift	4.43 (0.89) 4.24 (0.90)	0.19	0.17
Fatigue VAS							
Day 3 Post-shift Day 1 Pre-shift	4.1 (2.8) 3.5 (3.1)	0.62	0.14	Day 3 Post-shift Day 1 Pre-shift	4.5 (2.5) 2.5 (2.2)	2.0	0.001

Note: VAS, Visual Analog Scale. Morning samples were collected between 6 am–7 am, and evening samples between 6 pm–7 pm. Pre-shift designates the start of a shift; post-shift designates the end of the shift.

**Table 4 behavsci-13-00571-t004:** Correlations among study variables for day (n = 43) and night nurses (n = 38).

Variable	1.	2.	3.	4.	5.	6.	7.	8.
1. NSS	-	0.352 *	0.319 *	0.175	0.028	−0.112	0.158	0.107
2. PSS	0.228	-	0.354 *	0.171	−0.066	0.131	−0.021	0.259
3. Stress-VAS	0.332 *	0.230	-	0.273	−0.193	−0.122	0.219	0.433 *
4. MFI	0.383 *	−0.076	0.256	-	−0.222	−0.189	0.017	0.364 *
5. MSPSS	0.170	−0.155	0.000	−0.042	-	0.032	−0.016	−0.187
6. Day 3 Cortisol	−0.081	−0.022	−0.044	0.141	0.239	-	−0.014	0.147
7. Day 3 α-amylase	0.138	0.015	−0.039	0.143	0.333	0.118	-	0.203
8. Day 3 Fatigue	0.208	0.258	0.137	−0.005	−0.221	−0.126	0.137	-

Note: * *p* < 0.05. Day nurses are shown above the diagonal; night nurses are shown below the diagonal. NSS = Nurse Stress Scale. PSS = Perceived Stress Scale. Stress-VAS = overall stress scale. MFI = Multidimensional Fatigue Inventory. MSPSS = Multidimensional Scale of Perceived Social Support.

**Table 5 behavsci-13-00571-t005:** Hierarchical multiple regression predicting cortisol and alpha amylase from baseline personal (PSS) and work-related stress (NSS).

	Post-Shift Cortisol Day 3				Post-Shift Alpha Amylase Day 3			
	Day Shift		Night Shift		Day Shift		Night Shift	
Predictor	ΔR^2^	β	ΔR^2^	β	ΔR^2^	Β	ΔR^2^	β
Step 1	0.01		0.12		0.06		0.02	
Covariates								
Step 2	0.06		0.004		0.02		0.02	
Personal stress		−0.25		−0.06		−0.08		−0.05
Work stress		−0.01		−0.07		0.17		0.11
Total R^2^	0.07		0.12		0.08		0.04	

Note: Covariates were race, hours of sleep between shifts, and number of children in the home. Personal stress = PSS; Work Stress = NSS; General stress = VAS-S. Cortisol and alpha amylase are log-transformed. However none of the models were significant. Standardized beta was used. Tolerance levels for all regression models were checked for collinearity.

**Table 6 behavsci-13-00571-t006:** Hierarchical regression predicting fatigue from baseline personal (PSS), work-related (NSS), and overall stress (VAS).

	Post-Shift Fatigue Day 3		Post-Shift Fatigue Day 3	
Predictor	ΔR^2^	β	ΔR^2^	β
Step 1	0.17 *		0.19 *	
Covariates				
Step 2	0.02		0.08 **	
Perceived stress (PSS)		0.12		
Work stress (NSS)		0.08		
Overall stress (VAS-S)				0.30 **
Total R	0.19		0.27 **	

Note: Covariates were VAS-F, race, marital status, care for extended family, shift, and years of experience. VAS-S = Visual Analog Scale for Overall Stress at baseline; Post-Shift Fatigue = Visual Analog Scale Fatigue for overall fatigue at baseline. * *p* = 0.05; ** *p* = 0.01.

**Table 7 behavsci-13-00571-t007:** Moderation of social resources in the relationship between baseline stress and fatigue.

Predictor	ΔR^2^	β
Step 1 Covariates	0.27 *	
Step 2	0.02	
General stress (PSS)		0.15
Work stress (NSS)		0.02
Step 3	0.03	
Social Resources		
Step 4	0.06	−0.19
PSS × Social Resources		0.11
NSS × Social Resources		0.28 *
Total R	0.38	

Note: Covariates (race, marital status, care of extended family, care for extended family, years of experience, and VAS fatigue. NSS= Work-related stress; VAS-S = Visual Analog Scale for Overall Stress at baseline; VAS-F = Visual Analog Scale Fatigue for overall fatigue at baseline; MFI (Multidimensional Fatigue Inventory for baseline overall fatigue; MSPSS = Multidimensional Scale of Perceived Social Support for social resources. NSS and MSPSS were centered on avoiding multicollinearity. * *p* = 0.05.

## Data Availability

The data presented in this study are available on request from the corresponding author. The data are not publicly available due to privacy concerns.

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
