# Peer review of "Consecutive Shifts: A Repeated Measure Study to Evaluate Stress, Biomarkers, Social Support, and Fatigue in Medical/Surgical Nurses"

_behavsci, 2023, doi:10.3390/bs13070571_

Round 1

Reviewer 1 Report

I appreciate your submission of the scholarly work to BS. The paper demonstrates excellent writing and the authors have clearly invested significant effort. The chosen topic is highly relevant and carries significant implications. Implementing a repeated measures design can be challenging, but it provides relatively more accurate results. The theoretical foundation is robust, the methodology is sound, the analysis is conducted scientifically, and the discussion is well-presented. However, I would like to share a couple of observations.

1. Similarity index is 27% and 11% from a single source. As I do not know journal's plagiarism policy but it appears to be beyond tolerance zone.

2. For  the ease of readers, I suggest authors to create a separate section on theoretical and administrative implications of the findings after Discussion section. 

3. The manuscript carries typographic and spelling errors. For example see line 156. A careful proofreading is needed. 

Language is excellent. proofreading required

Author Response

Consecutive Shifts: A Repeated Measure Study to Evaluate Stress, Biomarkers, Social Support and Fatigue in Medical/Surgical Nurses

Reviewer comments in italics; responses in plain font.

Hello Reviewer 1,

Thank you for the opportunity to review this manuscript. I appreciate the difficulty of collecting multi-day biomarker data and also value the focus on systemic issues that impact employee health. Below I have a few questions/suggestions.

Reviewer 1 comments.

I appreciate your submission of the scholarly work to BS. The paper demonstrates excellent writing and the authors have clearly invested significant effort. The chosen topic is highly relevant and carries significant implications. Implementing a repeated measures design can be challenging, but it provides relatively more accurate results. The theoretical foundation is robust, the methodology is sound, the analysis is conducted scientifically, and the discussion is well-presented. share a couple of observations. 

  1. Similarity index is 27% and 11% from a single source. As I do not know the journal's plagiarism policy but it appears to be beyond the tolerance zone.

Thank you for raising this concern. We assume that the overlap is a function of the study being part of Dr. Cockerham’s dissertation. Moreover, the results were presented at various conferences: The Magnet Conference Atlanta, GA in 2019 and the Texas Medical Center Research and EBP Conference, Houston Texas in 2019. In addition, a pilot study that was conducted prior to the study reported in the manuscript, and topics were similar. This pilot study is cited in the manuscript, as follows: Cockerham, M., Kang, D.H., Howe, R., Weimer, S., Boss, L., & Kamat, S.R. Stress and Cortisol as Predictors of  Fatigue in Medical/Surgical Nurses and Nurse Leaders: A Biobehavioral Approach. J.NsgEduPrac, 2018, 8, 76-83. https://doi.org/10.5430/jnep.v8n5p76o. 

  1. For the ease of readers, I suggest authors create a separate section on the theoretical and administrative implications of the findings after the Discussion section. 

This section has been added to the Discussion section (second 4.3). 

3.The manuscript contains typographic and spelling errors. For example, see line 156. A careful proofreading is needed.  

I appreciate the feedback and the opportunity to make these changes. We have carefully proofed the paper and believe we have caught typographical errors and have worked to improve the manuscript's readability.

We thank you for your comments and believe the manuscript is much improved with your input.

Drs. Mona Cockerham & Margaret Beiers

Reviewer 2 Report

Thank you for the opportunity to review this manuscript. I appreciate the difficulty of colleting multi-day biomarker data, and also value the focus on systemic issues that impact employee health. Below I have a few questions/suggestions.

1. The results section was really difficult to follow. It often wasn't clear what statistic was being calculated or used to justify statements. It was unclear whether day and night nurses were being compared against each other or themselves, or both. So, just some tidying up of the analysis/results section to make it clear what is being done, and which hypothesis it's testing.

2. I would really appreciate a correlation matrix so the 1-1 relationships between variables can be more clearly seen and summarized (and to help with future meta-analyses).

3. It could just be how the manuscript was put into the format for reviewing, but Table 2 was really hard to follow, and section 1.4 came after hypotheses 4 which is the hypothesis that it's building up to. So tidying up of that can help.

4. The take aways actually weren't all that clear here, other than the value of social support. In fact, the moderation hypothesis/findings, which I think are the coolest, really got overshadowed by the rest of the study. So perhaps by really hammering those home up front it would help to better highlight them in the discussion.

I think it was really just a matter of how the manuscript was converted to this format for review, but in many instances there were words missing (see page 2 lines 61-87 for example) that made it quite hard to follow the manuscript at points.

Author Response

Reviewer comments in italics; responses in plain font.

Hello Reviewer, 2

We want to thank you for the opportunity to review this manuscript and for your feedback, it has made this a better manuscript. We are appreciative of the opportunity to share our work with your reading community. It is a foundational paper with the results of stress, biomarker and fatigue over three consecutive work shift providing a excellent foundation for future studies. Also highlighting the importance and significance of fatigue and social resources for nurses. Most of your comments were related to the statistical results sections. We believe the addition of several tables will increase the readability and understanding of our research.

We have left our track changes on our manuscript to show all changes since submission, and highlighted all the major changes.

Reviewer 2 comments.

  1. The results section was really difficult to follow. It often wasn't clear what statistic was being calculated or used to justify statements. It was unclear whether day and night nurses were being compared against each other or themselves, or both,. So, just some tidying up of the analysis/results section to make it clear what is being done, and which hypothesis it's testing.
  • What statistic was being calculated or used to justify statements?

Thank you for your overall comments and suggestions to more thoroughly describe the results section. Based on your feedback we have augmented the information in the results section and hope that we have clarified our approach: we have included a correlation table (Table 3) in addition to tables reporting the results of the hierarchical regression analyses conducted (Tables 4 - 6). We also walk through the analyses in more detail.

         Unclear whether day and night shift nurses are compared.

In addition to the changes cited above, we also are clearer in when we are comparing day and night nurses, and when we are combining data in one analysis. Because day and night nurses differed in their levels of cortisol, we could not combine them when examining the hypotheses including these biomarkers (something we now clearly state in the paper). But when we examine biomarkers, we examine day and night nurses separately. As such, for hypothesis 1 (which examined the increase in stress, biomarkers, and fatigue over time) we compared differences in day and night shift nurses. We also examine day and night nurses separately in the regression analysis for hypothesis 2 (which examined the relationship between baseline predictors of personal and work-related stress and levels of cortisol and α-amylase). Because there were no differences in any of the other assessments of stress and fatigue for day and night nurses, we combined them for examining our other hypotheses.

Comparison between nurses was within (each individual nurse) and between day and night   shift nurses, Line 193-194 under the Data Analysis section).

Yes, we compared within-person changes for day and night shift nurses separately in Table 2. We have tried to be clearer about our approach, and hope you find this section to be easier to follow.

Tidying up of the analysis/results section to make it clear what is being done, and which    hypothesis it's testing. See page 194-204 matched hypothesis and statistical analysis for clarity.

Thank you for this comment. We have also elaborated the statistical approach we use to match each hypothesis for clarity. We also think the addition of tables reporting our results will be helpful.

  1. I would really appreciate a correlation matrix so the 1-1 relationships between variables can be more clearly seen and summarized (and to help with future meta-analyses).

Correlations among all study variables are included in Table 3. Day shift nurse correlations are reported above the diagonal and night shift nurse data are reported below the diagonal.

  1. It could just be how the manuscript was put into the format for reviewing, but Table 2 was really hard to follow, and section 1.4 came after hypotheses 4 which is the hypothesis that it's building up to. So tidying up of that can help.

Thanks for catching this. We have made some formatting corrections to Table 2 to make it easier to follow. The ordering of the hypotheses and tables were difficult to read in the previous version of the paper. We have now reorganized the introduction to be clearer and more readable.

  1. The takeaways actually weren't all that clear here, other than the value of social support. In fact, the moderation hypothesis/findings, which I think are the coolest, really got overshadowed by the rest of the study. So perhaps by really hammering those home up front it would help to better highlight them in the discussion. Think it was really just a matter of how the manuscript was converted to this format for review, but in many instances, there were words missing (see page 2 lines 61-87 for example) that made it quite hard to follow the manuscript at points.

Thank you for this comment and for pushing us to articulate more the takeaway message of the study. We have tried to better articulate our findings in the Discussion section, particularly as related to social support. The results and discussion sections, along with the correction of line spacing and spelling, were corrected, and detail was given to improve the manuscript's clarity.

Many thanks,

Drs. Cockerham and Beier

 (Please note Dr. Kang passed away several years ago, but this was her last research project).
